Characterisation of the interaction of neuropilin-1 with heparin and a heparan sulfate mimetic library of heparin-derived sugars

Uniewicz Katarzyna A. 1
Ori Alessandro 2
Ahmed Yassir A.
Yates Edwin A.
Fernig David G. dgfernig@liv.ac.uk
Department of Biochemistry, Institute of Integrative Biology, University of Liverpool , Liverpool , United Kingdom
Uversky Vladimir
1 Current affiliation: Qiagen GmbH, Hilden, Germany

2 Current affiliation: EMBL Heidelberg, Heidelberg, Germany

Electronic publication date: 2014 Jun 26
Publication date: 2014
Volume: 2
Electronic Location ID: e461
Received 2014 Mar 31; Accepted 2014 Jun 9
Copyright: © 2014 Uniewicz et al.
Copyright year: 2014
Copyright holder: Uniewicz et al.
License: This is an open access article distributed under the terms of the Creative Commons Attribution License, which permits unrestricted use, distribution, reproduction and adaptation in any medium and for any purpose provided that it is properly attributed. For attribution, the original author(s), title, publication source (PeerJ) and either DOI or URL of the article must be cited.
License URL: https://creativecommons.org/licenses/by/4.0/

Keywords: Glycosaminoglycan, Heparin, Heparan sulfate, Neuropilin, Protein–polysaccharide interactions

Funding: European Commission North West Cancer Research Fund Cancer and Polio Research Fund The work was supported by the European Commission (Marie Curie Early Stage Training Fellowships to Alessandro Ori and Katarzyna Uniewicz), the North West Cancer Research Fund, and the Cancer and Polio Research Fund. The funders had no role in study design, data collection and analysis, decision to publish, or preparation of the manuscript.

==============================
Background. Neuropilin-1 (NRP-1) is a multidomain membrane protein with soluble isoforms interacting with a complex network of other membrane receptors, their respective ligands and heparan sulfate (HS). It is involved in the development of vasculature, neural patterning, immunological responses and pathological angiogenesis.

Methods. We have characterised the binding of a Fc fusion of rat NRP-1 (Fc rNRP-1) and of a soluble isoform, corresponding to the first four extracellular domains of human NRP-1, shNRP-1, using optical biosensor-based binding assays with a library of heparin derivatives. Selective labelling of lysines protected upon heparin binding allowed their identification by mass spectrometry.

Results. Fc rNRP-1 bound to heparin with high affinity (2.5 nM) and fast ka (9.8 × 106 M−1s−1). Unusually, NRP-1 bound both highly sulfated and completely desulfated stretches of heparin and exhibited a complex pattern of preferences for chemically modified heparins possessing one or two sulfate groups, e.g., it bound heparin with just a 6-O sulfate group better than heparin with any two of N-sulfate, 6-O sulfate and 2-O sulfate. Mass-spectrometry based mapping identified that, in addition to the expected the b1 domain, the a1, and c domains and the L2 linker were also involved in the interaction. In contrast, shNRP-1 bound heparin far more weakly. This could only be shown by affinity chromatography and by differential scanning fluorimetry.

Discussion. The results suggest that the interaction of NRP-1 with HS is more complex than anticipated and involving a far greater extent of the protein than just the b1–b2 domains. NRP-1’s preference for binding long saccharide structures suggests it has the potential to bind large segments of HS chains and so organise their local structure. In contrast, the four domain soluble isoform, shNRP-1 binds heparin weakly and so would be expected to diffuse away rapidly from the source cell.

Introduction

Neuropilin-1 (NRP-1, reviewed Uniewicz & Fernig, 2008), was first described in 1987 as antigen A5 expressed by neural cells of Xenopus tadpoles, with a role as a neuronal recognition molecule regulating neural path finding and cell differentiation. Subsequently, it was shown to play an important role in physiological and pathological angiogenesis (reviewed in Staton et al., 2007). NRP-1 has since been implicated in a wide range of functions ranging from immunological responses (Bruder et al., 2004), to cell adhesion through interaction with integrins (Fukasawa, Matsushita & Korc, 2007; Valdembri et al., 2009). NRP-1 is suggested to effect such varied cellular responses by virtue of its interactions with a wide range of functional partners. Soluble ligands include vascular endothelial growth factors (VEGF) (Makinen et al., 1999; Soker et al., 1997; Soker et al., 1998), fibroblast growth factors and hepatocyte growth factor/scatter factor (HGF/SF) (West et al., 2005) while receptors include those for VEGF (Fuh, Garcia & de Vos, 2000; Whitaker, Limberg & Rosenbaum, 2001) and for HGF/SF (MET) (Matsushita, Gotze & Korc, 2007). Moreover, NRP-1 can also interact with itself (Giger et al., 1998; Nakamura et al., 1998; West et al., 2005) and, as an added complexity, many of the protein partners of NRP-1 also bind HS (West et al., 2005) (reviewed Ori, Wilkinson & Fernig, 2008).

HS is a heterogeneous, polydisperse polysaccharide composed of repeating disaccharide units, comprising 1,4 linked uronic acid (α-L-iduronate, IdoA, or β-D-glucuronate, GlcA) and α-D-glucosamine (GlcN) with varying patterns of 2-O-sulfate in the former and 3-O-, 6-O-sulfate and N-sulfate or N-acetyl in the latter (reviewed Ori, Wilkinson & Fernig, 2008). HS also exhibits a domain structure, consisting of longer stretches of low sulfated disaccharide repeats, rich in D-GlcA (termed NA-domains), interspersed with shorter, more sulphated regions, in which higher levels of N-sulfated glucosamine and L-iduronate residues are found (termed S-domains) (Murphy et al., 2004). HS is a component of HS proteoglycans (HSPGs) present in the cellular membrane together with NRP-1, however, it is often substituted with heparin, its more fully sulfated experimental proxy, which is available in large quantities, owing to its role as a major anticoagulant therapeutic and resembles more closely, although is not identical to, the S-domains of HS.

X-ray crystallography has shown that NRP-1 interacts with heparin through the b1b2 domain, a region directly adjacent to the VEGF-A165 binding pocket (Appleton et al., 2007; Vander Kooi et al., 2007). NRP-1 has been proposed to dimerise in a heparin-dependent manner as a result of heparin and VEGF-A165 binding (Vander Kooi et al., 2007). Thus, one model of the angiogenic functional complex of NRP-1 comprises VEGF-A165: NRP-1: heparin at a 2:2:2 ratio, and the complex formed in this way would further interact with VEGFR-2 (Vander Kooi et al., 2007). In vitro studies have suggested that the addition of heparin enhances the binding of VEGF-A165 (Fuh, Garcia & de Vos, 2000; Gitay-Goren et al., 1992; Soker et al., 1996; Soker et al., 1998) and of placental growth factor-2 (Mamluk et al., 2002; Migdal et al., 1998) to NRP-1, and so elevate the number of binding sites for VEGF-A165 in a NRP-1 dependent manner, without affecting the affinity between NRP-1 and VEGF-A165 (Soker et al., 1996; Whitaker, Limberg & Rosenbaum, 2001). Moreover, binding of other growth factors, such as VEGF-C and VEGF-D, to NRP-1 appears to be fully dependent on the presence of heparin (Karpanen et al., 2006). However, other data point to a more complex relationship between these molecules. Thus, NRP-1 has been demonstrated to be a pure agonist of VEGFR2, indicating that it can stimulate angiogenesis independently of the VEGF-A ligand (Uniewicz, Cross & Fernig, 2010). Moreover, whereas heparin enhanced VEGF-A165 binding to NRP-1 and VEGFR-2, it also appeared to prevent NRP-1 interacting with VEGFR-1, suggesting a mechanism of functional competition that favours the formation of complexes of NRP-1 with VEGFR-1, or of NRP-1 with VEGFR-2, VEGF-A165 and heparin (Fuh, Garcia & de Vos, 2000). An important issue is that the characterisation of the functional role of the interaction of NRP-1 with HS has hitherto been restricted to structural studies using constructs encompassing just the b1b2 domains. Thus, the current structural work may not provide sufficient information to interpret the biological investigations.

To bridge more effectively biological and structural investigations the interaction of NRP-1 with HS/heparin is characterised here in detail, at the level of recognition of polysaccharide structural features by the protein and of the sugar binding sites in the protein. A recombinant dimeric Fc fusion of rat NRP-1 (Fc rNRP-1) was used to examine the interaction with heparin and a series of heparin derivatives. Fc rNRP-1 bound to heparin with high affinity (2.5 nM). The specificity of NRP-1 for structures in heparin were unusual. It interacted with both a persulfated and a fully desulfated heparin derivative. In stark contrast, human soluble monomeric NRP-1 (shNRP-1) exhibited much weaker interactions with heparin. The interactions of NRP-1 with heparin were further characterised using a “protect and label” approach (Ori et al., 2009). It was found that residues involved in heparin binding were present not just in the b1 domain, but also in the a1 and c domains, as well as in the L2 linker.

Experimental

Materials

Recombinant rat NRP-1 chimera (Fc rNRP-1), a soluble human truncated variant (shNRP-1) and human Fc were purchased from R&D Systems (Abingdon, Oxon, UK). The same heparin (17 kDa average molecular weight) (Celsus Lab, Cincinnati, OH, USA) was used throughout, including for the production of the chemically modified polysaccharides (Table 1) and oligosaccharides of defined size. Heparin derivatives were as characterised previously (Yates et al., 1996) and heparin derived oligosaccharides were obtained from partial heparinase I (IBEX Technologies, Montreal, Canada) digestion of porcine heparin performed, as described previously (Popplewell et al., 2009; Powell et al., 2010).

Table 1 Predominant structures and nomenclature of chemically modified heparins.

The letter I stands for iduronate, and A stands for the amino sugar glucosamine.

Analogue	Predominant repeat	IdoA-2	GlcN-6	GlcN-2	IdoA-3	GlcN-3*	
1(heparin)	I2SANS6S	SO3−	SO3−	SO3−	OH	OH	
2	I2SANAc6S	SO3−	SO3−	COCH3	OH	OH	
3	I2OHANS6S	OH	SO3−	SO3−	OH	OH	
4	I2SANS6OH	SO3−	OH	SO3−	OH	OH	
5	I2OHANAc6S	OH	SO3−	COCH3	OH	OH	
6	I2SANAc6OH	SO3−	OH	COCH3	OH	OH	
7	I2OHANS6OH	OH	OH	SO3−	OH	OH	
8	I2OHANAc6OH	OH	OH	COCH3	OH	OH	
9	I2S,3SA3S,NS6S	SO3−	SO3−	SO3−	SO3−	SO3−	
Notes.

* Numbers refer to the ring position of the carbon atoms.

Preparation of biosensor surfaces

Surfaces were prepared in an IAsys optical biosensor running IAsys plus software version 3.01. IAsys cuvettes functionalised with biotin (NeoSensors, Sedgefield, Cleveland, UK) were derivatised with streptavidin to enable subsequent capture of heparin that had been biotinylated at free amine groups (Rahmoune et al., 1998). Measurements were made at 20 °C in a final volume of 50 µL, unless indicated otherwise. Briefly, biotin functionalised cuvettes were incubated for 40 min with 20 µL 2.5 mg/mL streptavidin (Sigma-Aldrich, Gillingam, Kent, UK) in H2O and 10 µL of PBST (140 mM NaCl, 5 mM NaH2PO4, 5 mM Na2HPO4, 0.02% (v/v) Tween, 0.02% (w/v) sodium azide). After three washes in PBST the surface was incubated overnight at 4 °C with 20 µL, 10 mg/mL biotinylated heparin. The resonance scan confirmed that the surface was modified uniformly. The Fc rNRP-1 and Fc alone were found, as before (West et al., 2005), not to bind detectably to streptavidin derivatised surfaces.

Binding assays

The initial binding tests and the measurement of dissociation were performed in a manually operated IAsys biosensor; association data were acquired in an IAsys Auto Plus instrument. In kinetic association assays, 5 µL Fc rNRP-1 was injected into 45 µL of PBST after a stable baseline was acquired. Association data were collected for 240 s, after which the cuvette was washed with PBST and then regenerated by washing in 2 M NaCl, before returning the cuvette to 45 µL PBST for the following injection. Instrument noise was at most ±0.5 arc s.

For dissociation assays, higher concentrations of Fc rNRP-1 were used, 1.2 µg/mL in 50 µL of PBST, and heparin was injected as a competitor to reduce the probability of re-binding (Sadir, Forest & Lortat-Jacob, 1998). Following association and washing in PBST, dissociation was initiated by the injection of 1 µL 2 mg/mL heparin in PBST into the cuvette and followed for at least 60 s. Regeneration of the sensor surface was achieved as for the association assays with 2 M NaCl.

In competition assays 40 µL PBST and 5 µL competing sugar were added to the cuvette and a baseline was acquired. Fc rNRP-1 (5 µL, 5 µg/mL) was then injected and association data acquired for 300 s. The cuvette was then washed in PBST and regenerated with 2 M NaCl. As Fc rNRP-1 seemed to lose its heparin-binding activity over time, no more than 25 binding curves were collected from a single stock tube of the protein. At least two independent sets of data were acquired for each sugar.

Analysis of biosensor binding data

Data analysis was done using the FastFit software version 2.03 (NeoSensors) provided with the instrument. The validity of using a single site binding model to fit the data was established as before (West et al., 2005; Xu et al., 2012). Briefly, to allow a single site model to be used, data had to be randomly distributed around the model and in the case of association data, plots of the slope of initial rate and of kon against the concentration of NRP-1 had to be linear. The final kinetic values and their respective standard errors (SE) were calculated from 5 independent association experiments, and 6 independent dissociation experiments. In the figures, the response prior to the injection of protein is set to 0.

The competition assay data were normalised by comparison of extent of response of Fc rNRP-1 binding to the heparin surface versus the extent of Fc rNRP-1 response in the presence of sugar. The control extent of binding was defined as 100%, while values of Fc rNRP-1 binding in the presence of inhibiting compounds were calculated as percentages relative to this value. The IC50 values were calculated using Origin 8 (OriginLab Corporation, Northampton, MA, USA) applying a non-linear curve fit.

Selective labelling and identification of the heparin-binding peptides derived from Fc rNRP1

Mapping of the heparin-binding site was performed, as described previously (Ori et al., 2009), with some modifications. Briefly, to 20 µL of AF-heparin beads (Tosoh Biosciences GmbH, Stuttgart, Germany) packed in a minitip column and equilibrated in buffer A (17.6 mM Na2HPO4, 2.6 mM NaH2PO4, 100 mM NaCl, pH 7.6, 0.02% (v/v) Tween 20), 4 µg of Fc rNRP-1 was added in a total volume of 80 µL buffer A. The loading was repeated three times. After the column was washed with buffer A, amino groups not protected by the interaction with heparin were acetylated by the addition of 50 mM sulfo-NHS-acetate (Thermo, Pierce, Perbio Science, Cramlington, Northumberland, UK) in buffer B (18.3 mM Na2HPO4, 1.7 mM NaH2PO4, pH 7.8) for 5 min. The reaction was stopped by washing the column with buffer B. Acetylated proteins bound to the column were eluted with 2 M NaCl buffered with 45.75 mM Na2HPO4, 4.25 mM NaH2PO4, pH 7.8. The eluate was exchanged into this buffer (45.75 mM Na2HPO4, 4.25 mM NaH2PO4, pH 7.8) using a Vivaspin 100 kDa molecular weight cut-off centrifugal filter (Sartorius Stedim, Epsom, Surrey, UK) blocked previously with BSA to avoid Fc rNRP1 nonspecific binding to the filter. Subsequently, the lysines that were protected by heparin interaction were biotinylated using a final concentration of 10 mM NHS-biotin in dimethylsulfoxide (Thermo) and 30 min incubation at room temperature. The reaction was quenched by the addition of Tris–HCl pH 6.8 to a final concentration of 100 mM. Next, the sample was buffer-exchanged into 4.58 mM Na2HPO4, 0.43 mM NaH2PO4, pH 7.8 with a Vivaspin 100 kDa molecular weight cut-off centrifugal filter and dried by rotary evaporation.

The dried sample was dissolved in 8 M urea, 400 mM NH4HCO3 pH 7.8 and digested with 1.5 µg of chymotrypsin (Sigma-Aldrich), as previously described (Ori et al., 2009). Biotinylated peptides were enriched using Strep-Tactin Sepharose (IBA, Stratech, Newmarket, Suffolk, UK) minitip columns (Ori et al., 2009) and desalted using C18 ZipTip™ (Millipore, Watford, Herts, UK) according to the manufacturer’s instruction.

Peptide analysis

Desalted peptides were dissolved in 20 µl of 0.1% (v/v) formic acid in HPLC-grade water and directly infused into a quadrupole-time of flight (Q-TOF) mass spectrometer (Waters Corporation) at 0.2 µl/min. The mass spectrometer was operated in the positive ion nano electrospray mode with a source temperature of 80 °C and capillary voltage 2.8 kV. First, a survey scan was performed in MS-TOF mode in the range between 200 and 2,000 m/z. The most intense ions were then manually selected and subjected to MS/MS analysis. For each ion the MS/MS spectrum was acquired in the MS-Q-TOF mode. The precursor ion was selected in the quadrupole, fragmented by collision-induced dissociation and then fragment ions were recorded by TOF (range: 80–3,000 m/z). The collision gas was argon and the collision energy was manually adjusted during the acquisition between 25 and 45 V. Peak lists were extracted using the MaxEnt3 algorithm applying a 5% base peak intensity cut-off. Data analysis was performed using the MS-tag tool of the Protein Prospector package v.5.2.2 (http://prospector2.ucsf.edu) using the following parameters: digest = chymotrypsin; max missed cleavages = 5; possible modifications = acetyl (Lys), biotin (Lys), carbamidomethyl (Cys), carboxymethyl (Cys); parental ion tolerance = 100 ppm; fragment ion tolerance = 300 ppm; non-specific cleavage = at 1 termini. Both carbamidomethylation and carboxymethylation of cysteines were included as variable modifications since S-carbamidomethyl-cysteine can be hydrolysed to S-carboxymethyl-cysteine under the acid condition used for the elution of biotinylated peptides from Strep-Tactin beads. The analysis was performed for monoisotopic ions with an error tolerance of 100 ppm for parental ions, and 300 ppm for fragmented ions. The UniProt accession numbers of the proteins analysed were used as a pre-search parameter. All other settings were used as default values.

Homology modelling of human NRP-1

The sequence of NRP-1 (Uniprot accession number O14786, residues 5-564) was homology-modelled based on the 2QQM and 2QQK structures (Appleton et al., 2007), and the a1 and c domains modelled independently applying the same approach. The consistency of the secondary structures within the templates was verified by Dalilite (Holm & Park, 2000). The alignment was performed by Clustal W (Thompson, Higgins & Gibson, 1994) and its PIR presentation served to prepare the input file for Modeller (Eswar et al., 2008). Subsequent modelling produced five initial structures, which were verified by DOPE score in Modeller 9v7, and the lowest score model (most energetically favoured) was further optimised. In order to do so, the Ramachandran plot of the chosen model was examined, and residues in the outlier region were remodelled to obtain correct biophysical properties. Again, the best model was validated by the DOPE score, and additionally inspected by Model Quality Assessment Suite, and compared with the template structural files. This showed that the obtained model was of medium quality, however, at the same time it did not differ substantially in its properties from the input templates and it had no major structural constraints. The final model was used to visualise the peptides identified in the selective labelling experiment. Alignments were performed by Clustal W (Thompson, Higgins & Gibson, 1994) and served to reflect best fitting of the identified peptides on the available crystal structure. In order to present location of the identified biotinylated peptides of rat NRP-1, the homology-modelled human NRP-1 was used. Subsequently, Pymol was used to illustrate these regions.

Dual polarisation interferometry (DPI)

DPI analysis was performed on an AnaLight Bio200 System (Farfield Scientific, Manchester, UK). The experiment was performed as described previously (Popplewell et al., 2009). Briefly, a thiol chip was derivatized at 30 °C with 5 mg/mL N-(β-maleimidopropionic acid) hydrazide (BMPH) (Thermo) applied at 8 µL/ min to each flow cell to produce a surface with hydrazide groups. Following washing with PBS at 50 µL/min, the flow cell destined to be modified with oligosaccharide was injected with 180 µL of 2 mg/mL heparin-derived hexadecasaccharide (dp 16) dissolved in PBS pH 5 at 2 µL/min. Subsequently, both flow cells were reacted with aldehyde-functionalised poly(ethylene glycol) (PEG-CHO, 750 Da) (Rapp Polymere, Tübingen, Germany) to block unreacted hydrazide groups. PEG-CHO (180 µL of 20 mg/mL) dissolved in PBS was applied at 1 µL/min and flow was stopped to allow an overnight incubation. The injection of PEG-CHO was then repeated. After extensive PBS washes the temperature was returned to 20 °C and the surface was washed with 2 M NaCl in 10 mM NaH2PO4 pH 7.5.

For binding assays proteins were added at 3.33 µg/mL (14 nM Fc NRP-1, 47 nM hNRP-1 and 123 control Fc) at 50 µL/min. After each binding assay, sensor surfaces were regenerated by a 30 s wash with 20 mM HCl at 50 µL/min and then returned to PBS.

DPI data analysis

The derivatization of the surface and the subsequent protein-binding reactions were analysed with Analight Explorer according to the manufacturer’s instructions. The mass readings of the binding are from the baseline prior to injection of protein, which was set to 0.

Differential scanning fluorimetry (DSF)

DSF was performed as described previously (Uniewicz et al., 2010). Briefly, 0.57 µM (136 µg/mL) Fc rNRP1 was tested alone or with 1.132 µM (19 µg/ mL) heparin or 11.32 µM (190 µg/mL) heparin, to give 1:2 and 1:20 molar ratios of dimeric Fc rNRP1 to heparin, respectively. Similarly, a final concentration of 1.132 µM shNRP1 was tested with 1.132 µM heparin or 11.32 µM heparin (1:1 and 1:10 monomeric truncated shNRP1 to heparin, respectively). The assay was performed in 96-well Reaction Plates (Applied Biosystems, Warrington, Cheshire, UK). The final volume (25 µL) comprised the protein dissolved in PBS (20% v/v), heparin dissolved in HPLC grade water (10% v/v), PBS (60% v/v or 70% v/v in case of the condition without heparin) and a freshly prepared 100× water based dilution of Sypro Orange 5000× (Invitrogen) (10% v/v), added in the given order. During the additions, the plate was kept on ice and immediately afterwards it was sealed with Optical Adhesive Film (Applied Biosystems) and directly analysed in a 7500 Fast Real Time PCR System (software version 1.4.0) (Applied Biosystems) instrument. The heating cycle comprised a 120 s pre-warming step to 31 °C and a subsequent gradient between 32 and 81 °C obtained in 99 steps of 20 s, each of 0.5 °C. Data were collected using the calibration setting for TAMRA™ dye detection (λex 560 nm; λ em 582 nm) installed on the instrument (as compared to Sypro Orange ideal settings λ ex 492 nm; λ em 610 nm). The data were analysed using the Plot v. 0.997 software for Mac OS X by application of an exponential correlation function approximation of the first derivative for each melting curve.

Results and Discussion

The commercially available NRP-1 used in this study, Fc rNRP-1, encompasses most of the extracellular part of the native protein with the omission of amino acid residues 811–828. The entire C-terminal transmembrane/ intracellular region is absent. The protein is a chimeric molecule, in which the rat NRP-1 sequence is fused to a Xa factor cleavage site, followed by the Fc part of human IgG1 sequence, where the latter has the ability to form disulfide bridges between the Fc domains (Figs. S1A and S1B). Consequently, the final protein is a dimer. Such an approach, where proteins are expressed as dimers, is often chosen to increase the stability of the final product. Consequently, excluding the small deletion, the assayed protein comprises most of the extracellular sequence of NRP-1 and is a soluble dimer. The protein also possesses a C-terminal hexahistidine tag.

Fc rNRP-1 is modified by N-glycosylation, as shown by the increased electrophoretic mobility following treatment with N-glycanase (Fig. S1D); the Fc part of Fc rNRP-1 is also the subject of N-glycosylation (Fig. S1D). Additionally, analysis under native conditions revealed that the whole protein migrates as a large aggregate, substantially exceeding the estimated size of the intact dimer (Fig. S1E). This observation is in accord with previously observed oligomerisation of Fc rNRP-1 in gel filtration (West et al., 2005) and implies that the protein in solution is an oligomer.

Characterisation of kinetics of the Fc rNRP-1 interaction with heparin by optical biosensor binding assays

The interaction of Fc rNRP-1 with heparin was analysed in an optical biosensor. Heparin is a polymeric ligand, possessing a range of binding sites and immobilisation of the heparin to the surface through streptavidin may result in steric hindrance of some of these sites. Thus, at higher concentrations of ligate, secondary binding sites can appear in the analysis (Fernig, 2001). As a consequence, a concentration range of Fc rNRP-1 of 0.25–1.22 nM was used, as only within this range was the interaction characterised by a one-site binding model. This means that, as with all such analyses, it is the interaction of Fc rNRP-1 with a class of highest affinity sites within heparin that is being measured, not the average of all the potential binding sites in heparin. Increasing concentrations of Fc rNRP-1 showed increasing extents of binding (Fig. 1A). Each concentration exhibited a one-site binding model, since the data were distributed randomly around the model (Figs. 1B–1F). The initial rate of binding, defined by the binding of the Fc NRP-1 within the first 15 s of its addition was shown to increase in a linear manner with increasing concentration (Fig. 1G). This illustrated that the binding of Fc rNRP-1 to heparin was not limited by mass transport or by steric hindrance (Edwards et al., 1995). The calculated on-rate (kon) was observed to increase linearly with the concentration of Fc rNRP-1, which again supported the use of a one site model to analyse the data (Fig. 1H). kdiss, was measured directly in a separate set of dissociation experiments, where the addition of competing heparin in solution was used to prevent the re-binding of Fc rNRP-1 to the heparin surface, a common artefact in such surface measurements (Sadir, Forest & Lortat-Jacob, 1998). ka was extremely fast, 9.8 × 106 M−1 s−1 (Table 2). Of the protein-GAG interactions that have been studied, only that of HGF/SF with dermatan sulfate has been found to be faster (Lyon et al., 1998). With a kd of 0.025 s−1, the Kd of the interaction is of high affinity (2.5 nM). The measured kd is considerably faster than that calculated from the intercepts of the plots of a konversus concentration (Table 2). This reflects the absence of re-binding artefacts owing to the use of competing heparin during the dissociation phase in the direct measurements of kd and is one explanation why a previous study (Narazaki, Segarra & Tosato, 2008) determined a slower kd, similar to that derived here from the intercept. Alternatively, the use of just the b1–b2 domain in the latter study may explain the slower kd (Narazaki, Segarra & Tosato, 2008). Clearly, Fc rNRP-1 can be classified among the high affinity interacting partners of heparin, though it must be stressed that this is a measurement for the highest affinity binding sites in the sugar and it is likely that there will be many lower affinity sites.

Figure 1 Kinetic analysis of Fc rNRP-1 binding to a heparin-derivatised surface in an IAsys optical biosensor.

Data shown are a representative of a set of five independent experiments described in the experimental section. (A) Extent of binding of indicated molar amounts of Fc rNRP-1 (arc s). (B–F) The distribution of the data points (jagged line) around a one site binding model (horizontal line at 0 arc s) is shown for each of the concentrations of Fc rNRP-1 used in the binding assay in panel A. (B) 0.25 nM. (C) 0.33 nM. (D) 0.55 nM. (E) 0.94 nM. (F) 1.22 nM. (G) Linear relationship between the slope of initial rate of association and concentration of Fc rNRP-1. (H) Linear relationship between kon, determined from a one-site model, and concentration of Fc rNRP-1.

Table 2 Kinetics and affinity of Fc rNRP-1 binding to a heparin-derivatised surface in an optical biosensor.

ka is the association rate constant, p value is the correlation coefficient of the linear regression through the kon values, intercept is the kd value obtained from kon plots , kd is the dissociation rate constant obtained in the dissociation experiments in the presence of competing heparin, and Kd is the affinity calculated from the ratio of kd/ka. The standard errors (SE) were calculated from five independent association datasets and six independent dissociation datasets and combined for the calculation of Kd.

ka (Ms)−1	p	Intercept (s−1)	kd (s−1)	Kd (nM)	
9,800,000 ± 2,100,000	0.96	0.005 ± 0.0001	0.025 ± 0.0007	2.5±0.5	

Analysis of the structural requirements of Fc rNRP-1 binding to heparin

After characterisation of the kinetics of the interaction of Fc rNRP-1 with heparin, the next question focused on the structural requirements of heparin that favour this interaction. Two different parameters were examined, namely, the length of the heparin oligosaccharide and the sulfation pattern of heparin. While the first of these describes the architectural attribute of the best fitted structure within the binding site of the Fc rNRP-1, the latter may provide additional functional information, as different sulfation patterns have distinct solution conformations and so functional properties (Rudd et al., 2007).

The identification of the binding preferences was established in competition tests, where Fc rNRP-1 was compelled to select between binding to the heparin immobilised on the sensor surface and a heparin-derived compound present in solution. In this way, the capacity of the assayed compounds to compete for Fc rNRP-1 binding to the heparin surface can be compared with the same capacity of heparin added to solution.

The analysis of the minimal length of oligosaccharide that can be accommodated by Fc rNRP-1 applied compounds defined by their degree of polymerisation (dp), where dp 2 is the minimal repeating disaccharide unit of heparin. The range used in this study covered dp 2–dp 26. Oligosaccharides of dp 2–dp 8 had very weak competing potency, however, starting from dp 16, a substantial inhibition of binding of the protein to the surface was observed (Figs. 2A, 2B and S2). It is noteworthy that the largest oligosaccharide tested, dp 26, was still significantly less effective than heparin (Figs. 2A, 2B and S2), which corresponded to the difference in size between dp 26 and heparin (16 kDa, ∼dp 34). Thus, while there was a clear correlation between the size of an oligosaccharide and its binding capability, there would appear to be a more complex relation between the interaction of Fc rNRP-1 and of heparin, as full inhibition could only be observed with native heparin (Figs. 2A, 2B and S2). This preference for heparin, which is polydisperse in length and sulfation, may reflect the size of the sugar binding site in the Fc rNRP-1, which may be increased by the dimeric structure of Fc fusion protein or the requirement for a structure for the highest affinity binding that is more common in the longest chains of heparin.

Figure 2 Competition experiments defining structural requirements of heparin derivatives that enable Fc rNRP-1 binding.

Inhibiting ability of oligosaccharides of selected lengths and modified heparin derivatives on Fc rNRP-1 binding to the heparin derivatised surface was tested, as described in the experimental section. (A), (C) The relative binding values at each concentration of added sugar competitor were adjusted to 100% of Fc rNRP-1 binding alone to the immobilised heparin (measured at between 10–40 arc s). In (A) only a selection of curves demonstrating the largest changes in IC50 are shown for clarity, the full set are in Fig. S2). The experiment was performed three times independently (mean ± SD). (B), (D) IC50 values of the compounds from the experiments described in panels A and C were calculated from the non-linear curve fit of the respective binding curves and are expressed in (mean ± SE µg/mL).

Next, the potency of variously sulfated heparin derivatives was tested. The compounds were chemically produced from native heparin and enable an evaluation of the importance of particular patterns of sulfation for Fc rNRP-1 binding. Removal of one sulfate group from heparin (Table 1, Figs. 2C and 2D; NAc, 2-OH, 6-OH) caused a substantial reduction in binding (IC50 increased by 40- to 77-fold), with Fc NRP-1 showing a graded preference for heparin containing 2-O sulfate and N-sulfate groups over 6-O sulfate groups. Intriguingly, these preferences did not extend to heparins lacking two of three sulfate groups. Thus, whilst heparin with just N-sulfate (2-OH, 6-OH) and 2-O sulfate (NAc, 6-OH) had IC50 values 65- and 118-fold higher than native heparin, respectively, heparin with just 6-O sulfate groups (NAc, 2-OH) had an IC50 lower than any of the monodesulfated heparins (Figs. 2C and 2D). Moreover, Fc NRP1 bound completely desulfated heparin at least as well as the monodesulfated heparin and far better than heparin carrying just 2-O sulfate or N-sulfate groups. However, persulfated heparin was the most effective competitor, with an IC50 lower than that of heparin.

These data indicate that the binding motif of Fc rNRP-1 in heparin is more complex than one described by a simply linear sequence of sulfated sugar residues or by simple ion-exchange. Thus, it would seem that although ionic interactions are important, there are other interactions, which allow a sugar structure to bind to Fc rNRP-1 quite effectively. Moreover, persulfation places sulfate groups at the C3 position of glucosamine and it may be that this is an important requirement for Fc rNRP-1 binding. This is intriguing, since 3-O sulfated structures are rare in heparin and cellular HS (Lindahl, Kusche-Gullberg & Kjellen, 1998).

Identification of the protected regions of Fc rNRP-1 upon interaction with heparin

A structural proteomics approach, termed “protect and label” (Ori et al., 2009), was used to identify heparin binding sites in Fc rNRP-1. This method has been applied successfully to identify heparin-binding sites in single domain proteins, such as a series of FGFs, platelet factor 4 and pleiotrophin (Ori et al., 2009; Xu et al., 2012). Here it was used for the first time to study a multidomain protein, Fc rNRP-1.

In initial experiments, it was found that while Fc rNRP-1 bound the heparin column effectively, it was released during the acetylation step. One explanation might be that, although Fc rNRP-1 binds large oligosaccharides with high affinity, the interaction might be quite dynamic at the atomic level, such that the protein might “rock” on its polysaccharide-binding site(s). In the presence of NHS acetate, as lysines involved in heparin binding became temporarily exposed they would become acetylated and consequently not be able to re-bind the protein. Alternatively, the reversal of charge on many exposed lysines not involved in heparin binding may alter the conformation of the protein so that it binds heparin less effectively. This would be consistent with the known effects of chemical modification of lysines in some proteins, namely structural rearrangement (Nakagawa, Capetillo & Jirgensons, 1972) and loss of stability (Fazili, Mir & Qasim, 1993). Therefore, the initial binding step was performed in PBS, but the subsequent acetylation step was done without NaCl, to prevent loss of Fc rNRP-1 (Fig. 3A).

Figure 3 Identification of the protected regions of Fc rNRP-1 upon interaction with heparin.

(A) Represents the steps of analysis of Fc rNRP-1. Lane (LD) corresponds to the sample applied to the heparin column, lane (FT) is the flow through or unbound material after applying the sample three times to the column, lanes (B1 and B2) are the flow through from the two acetylation steps, lane (W1) is the post acetylation wash of the column, elution (E1) is the material eluted from the column with 2 M NaCl, lane (E2) is the same material after biotinylation (E2), and (E3) is the biotinylated material after concentration and buffer exchange. Five% (v/v) of the total material (4 µg) was analysed on SDS-PAGE and silver stained. (B) Schematic structure of Fc rNRP-1, where peptides with biotinylated lysines were identified are labelled as black squares. (C) Two perspectives of the NRP-1 structure based on in silico modelling (a1–b2 domains) with highlighted residues corresponding to the assigned peptides according to the sequence alignment. In green labelled regions identified by the Protect and Label strategy, in blue the residues identified as heparin-interacting according to Vander Kooi et al. (2007). See Fig. S3 for the alignment of NRP-1 sequences corresponding to this structure, with annotated identified peptides.

This allowed the generation of 15 ions that were assigned to particular sequences (Table 3, Fig. S5). Five other ions were not matched to any region of the recombinant Fc NRP-1, which might be due to modifications (either posttranslational, e.g., glycosylation, or due to the experiment) occurring, which were not accounted for. All assigned ions encompassed one biotinylated lysine residue, and corresponded to the NRP-1 moiety of the fusion protein or the IgG1 Fc moiety (Table 3). The peptides identified within NRP-1 moiety were mapped to the a1, b1, c domains and to the linker following the c domain (L2) (Table 3). Within the a1 domain two sequences were identified based on three independent ions, whereas within the b1 domain one largely overlapping sequence was identified by two independent ions. Within the c domain three sequences were identified from one ion each, while the fragment within the L2 region was identified by two independent ions (Table 3). In the IgG1 Fc part of the recombinant protein four peptide sequences were identified, where one of them were based on two independent ions, and the remaining three on one ion each. The list of peaks with assigned values is described in Fig. S5.

Table 3 List of peptides identified in the protect and label analysis of the binding sites for heparin in Fc rNRP-1.

The complete list of ions identified by MALDI-Q-TOF mass spectrometery is presented according to the descending intensity of the ions, along with their observed and theoretical masses with appropriate error values (expressed in ppm) and directly linked to the recognised sequences and their respective location within the Fc rNRP-1. The biotinylated lysines are indicated in bold. Assigned spectra are available in the Supplementary data (Fig. S5).

Peptide	Mass experimental	Mass calculated	Error (ppm)	Sequence	Residues	Domain	
1	1800.96	1800.91	24.3	VRIK(Biotin)PASWETGISM	404–417	b1	
2	1494.76	1494.6464	76	AGAFR(Acetyl)SDK(Biotin)C(Carbamidomethyl)GGT	19–30	a1	
3	1926.02	1926.0953	-39.1	DK(Biotin)NISRKPGNVLKTL(Ammonium)	822–836	L2	
4	2024.1	2024.0481	26.4	TLPPSRDELTK(Biotin)NQVSL	978–993	IgG1 Fc	
5	2125.14	2125.0059	63.1	VVVDVSHEDPEVK(Biotin)FNW	890–905	IgG1 Fc	
6	1452.74	1452.6358	68.5	AGAFRSDK(Biotin)C(Carbamidomethyl)GGT	19–30	a1	
7	1790.91	1790.8643	26.1	TVDK(Biotin)SRWQQGNVF	1039–1051	IgG1 Fc	
8	1381.71	1381.6603	43.3	NGK(Biotin)EYKCKVS	943–951	IgG1 Fc	
9	1516.72	1516.7464	−14.7	QVIFEGEIGK(Biotin)GN	778–789	c	
10	1811.92	1811.9069	7.25	SQADENQK(Biotin)GK(Acetyl)VARL	695-708	c	
11	2074.06	2074.0109	28	MVVGHQGDHWK(Biotin)EGRVL	754–769	c	
12	1182.65	1182.6452	4.05	VRIK(Biotin)PASW	404–411	b1	
13	1909	1909.0688	−36	DK(Biotin)NISRKPGNVLKTL	822–836	L2	
14	1628.7	1628.6621	23.3	HSYHPSEK(Biotin)CEW	46–56	a1	
15	1364.69	1364.6337	41.3	NGK(Biotin)EYKCKVS -NH3	943–951	IgG1 Fc	

These data identify part of the known heparin-binding region of NRP-1 in the b1–b2 domains (Fig. 3, blue label, Fig. S3) (Mamluk et al., 2002) and, importantly, extend heparin binding to the a1, c domains and the L2 linker (Figs. 3B, 3C, and S3) . The identification of regions within the IgG1 Fc part of the recombinant protein was surprising, as in optical biosensor and DSF measurements the IgG1 Fc did not interact with heparin or heparin-derived oligosaccharides (Figs. 4 and 5 and West et al., 2005). This may be the consequence of Fc possessing weak heparin binding sites (∼mM affinity), which only become apparent when it is placed in proximity to heparin on the affinity column by virtue of the primary interaction ofNRP-1. Similar arguments may contribute to the identification of lysine residues in the NRP-1 moiety outside the b1–b2 domains as interacting with heparin. Several lines of evidence support the argument that the novel binding sites in rNRP-1 represent real sites of interaction. The main structural study of the residues involved in heparin interaction is based only on the b1b2 domain construct (Vander Kooi et al., 2007), so it could not predict other areas of interaction. Moreover, secondary binding sites, which may be of lower affinity, as found in FGFs (Ori et al., 2009; Xu et al., 2012) may have been overlooked in favour of a dominant high affinity site. Thus, the current model of the interaction of NRP-1 with heparin describes specifically the interaction with the b1–b2 domains, but not the whole protein, and so may lose the context of the additional domains present in NRP-1. Analysis of the conservation of the lysines identified in Fc-rNRP-1 across vertebrates shows that most are conserved (Fig. S3). In two cases (K22 and K 746) the zebrafish has a basic residue one amino acid upstream and only for K832 in Fc-rNRP-1 is there no corresponding basic residue in chicken, Xenopus laevis or zebrafish. The conservation of these residues is consistent with their structural or functional importance, which may relate to their involvement in binding to HS.

Figure 4 Comparison of polysaccharide-binding properties of Fc rNRP-1, shNRP-1 and Fc.

Binding of Fc rNRP-1, shNRP-1and Fc to (A) a heparin-derivatised surface in an IAsys optical biosensor and to (B) a dp 16 derivatised surface in a dual polarisation interferometer.

Figure 5 Sypro stability assay of recombinant NRP-1s in the presence and absence of heparin.

(A) Fc rNRP-1, (B) shNRP-1 and (C) Fc were subjected to denaturation cycle alone or with the indicated amount of heparin in the presence of Sypro Orange dye. The melting curves (mean of three shown) were recorded by the RT-PCR instrument and analysed as described in the Methods. The first derivatives of the melting curves were plotted. (D) The melting temperature (Tm) values equal to the maxima of the derivatives from A, B, C.

To visualise the identified peptides on a 3-D structural model the most complete available structures of NRP proteins were selected for the in silico modelling purpose. The structures of NRP proteins, 2QQM and 2QQK, cover only a1a2b1b2 domains of NRP-1 and NRP-2 (Appleton et al., 2007). This region of the protein is followed by long unstructured elements, namely the L1 and L2 linkers, and the c domain, and to date no data are available on their possible conformation, as such regions are problematic for structural studies. Similarly, owing to the substantial size of these elements no in silico approach is capable of making a prediction of their conformation. The appropriate residues were highlighted within the modelled structure (Fig. 3C). Only five of the peptides identified in the Protect and Label experiment could be mapped on the available structure, and are indicated by 3 regions highlighted in green. Three of them mapped to the a1 domain, while other two were found to be in close proximity of the heparin-binding site in the b2 domain identified in other experiments, which is shown in blue (Vander Kooi et al., 2007). Unfortunately, the remaining five peptides were not covered by the structure, since they were localised in the c domain and L2 linker following the c domain.

The interaction of shNRP-1 with heparin

Given the differences found in the interaction of Fc rNRP-1 with heparin compared to that of the b1–b2 domain, we investigated the properties of shNRP-1 (Fig. S1). shNRP-1 covers the sequence of a naturally occurring alternatively spliced soluble human isoform of NRP-1 and corresponds to domains a1, a2, b1, b2, but lacks the L2 linker and the c domain (Fig. S1). Thus, it contains the b1–b2 domain that covers the hitherto established heparin binding site of NRP-1. shNRP-1 was found to be modified by N-glycosylation, though in contrast to Fc rNRP-1, in native PAGE conditions it appeared to migrate as a monomer (Fig. S1).

Surprisingly, initial experiments revealed that there was no detectable binding of shNRP-1 to heparin-derivatised surfaces in optical biosensor experiments (Fig. 4). Here, heparin was immobilised to the surface by means of biotin groups introduced to its free amino groups. It has previously been shown in the case of cyclophilin B, that such internal biotinylation can abrogate the interaction of a protein with heparin (Vanpouille et al., 2007). To exclude the possibility that biotinylation of heparin was interfering with shNRP-1 binding, a dp 16 was immobilised through biotin at its reducing end onto the more sensitive dual polarisation interferometer (West et al., 2005). Again, the binding of shNRP-1 was not detectable (Fig. 4). However, shNRP-1 was observed to bind to Toyopearl heparin resin, indicating that the protein probably does indeed bind heparin. Although shNRP-1 bound to the heparin column, it was observed to consistently elute from the column during the post-acetylation wash steps, despite these being of low ionic strength (Fig. S4), precluding any further analysis.

Overall, these data support the argument that shNRP-1 interacts with heparin much more weakly than Fc NRP1 (this work) or the b1–b2 domains in isolation (Narazaki, Segarra & Tosato, 2008). To provide further evidence for this interaction, the thermostabilisation of proteins that follows their binding to heparin was exploited (Uniewicz et al., 2010). To establish comparable conditions for the analysis of recombinant NRP-1s, it was assumed that each NRP-1 moiety had the potential to bind to one heparin molecule, therefore, whereas shNRP-1 was assayed at ratios of heparin of 1:1 and 1:10, Fc rNRP-1 was assayed at 1:2 and 1:20. Both isoforms displayed higher melting temperatures (Tm values) in the presence of heparin, which supported the capability of shNRP-1 to bind to heparin (Fig. 5). Again, the Fc part alone was confirmed to not interact with heparin (Fig. 5).

Implications of the interactions of Fc rNRP-1 and shNRP-1 with heparin and the heparin-derived sugar library

We have extended the characterisation of the interaction of NRP-1 with heparin/HS from just the b1b2 domain to the entire protein in two different contexts: as an Fc fusion, Fc rNRP-1 and as a monomer corresponding to a naturally occurring soluble isoform covering the first four extracellular domains, shNRP-1. There are clear physical differences in the two proteins. The Fc rNRP-1 fusion will be a dimer and this has a clear tendency self-associate into oligomers (Fig. S1 and West et al., 2005), whereas shNRP-1 remains a monomer (Fig. S1). A likely reason for the difference in the state of oligomerisation of the two proteins is the MAM type c domain and these are often found to drive protein-protein interactions (reviewed Uniewicz & Fernig, 2008). Thus, although there are no data to support this directly, it is possible that the c domain drives self-association of NRP-1 in the membrane, which would be enhanced by the reduced dimensionality afforded by this cellular location. Consequently, the Fc rNRP-1 fusion protein may in this respect be a reasonable model for membrane NRP-1.

Analysis of the interaction of Fc rNRP-1 with heparin identified some unusual features. The interaction is of high affinity with a high association rate constant. In terms of binding sites in the sugar, Fc rNRP-1 clearly has a preference for longer structures, alongside an ability to bind both highly suflated and non-sulfated structures (Fig. 2). The identification of several distinct sites of interaction in Fc NRP-1 in the protect and label experiments (Table 3) is consistent with the preference for long sequences of heparin. While it is possible that the peptides identified in these experiments in the a1 and c domains and in the L2 linker were an artefact of the low ionic strength required for the protection step, it should be noted that only a proportion of lysine residues in Fc rNRP-1 were labelled. This argues that by binding Fc NRP-1 to heparin in PBS the specificity of interactions was maintained and suggests that NRP-1 may indeed have multiple heparin binding sites. Considering the size of NRP-1, this seems a reasonable interpretation, since smaller, single domain proteins such as endostatin, RANTES (Sasaki et al., 1999; Shaw et al., 2004) and members of the FGF family (Ori et al., 2009; Xu et al., 2012) have well documented multiple heparin binding sites. Morever, these sites seem to be conserved across evolution in the case of the FGFs (Xu et al., 2012) and NRP-1 (Fig. S3). Given the large size of the most effective heparin-derived binding structures and the observation that heparin and desulfated heparin bind competitively, this further suggests that polysaccharides of different structure may engage overlapping sites in the rFc NRP-1. However, it is not possible to distinguish protection of a lysine residue by heparin from protection by an intramolecular interaction; that the latter may in some instances be caused by conformational change/domain re-arrangement triggered by heparin binding is clearly possible, since heparin bound Fc NRP-1 has a substantially increased melting temperature (Fig. 5), indicative of structural change in the protein. Thus, which of the peptides encompassing protected and labelled lysine residues are directly involved in heparin binding remains to be established directly.

HS, which will be the physiological polysaccharide NRP-1 binds in the pericellular matrix of cells, is less sulfated than heparin and it has a domain architecture (Murphy et al., 2004). Thus, in HS sulfated tracts of saccharides (S-domains) are relatively short, flanked by transition domains (NAS domains, lower sulfation with ∼1 glucosamine in two N-sulfated), separated by non sulfated NA domains of repeating disaccharides of glucuronic acid and N-acetyl glucosamine. Although the interaction of Fc rNRP-1 with desulfated heparin suggests that it would bind the NA domains, the disaccharide units of the latter contain glucuronic acid residues, whereas in desulfated heparin iduronic acid will be the dominant uronic acid epimer. The flexibility of the iduronate acid ring (Torri et al., 1985) is recognised to be important in protein binding (reviewed Ori, Wilkinson & Fernig, 2008). Thus, the present data suggest two possible modes of binding of NRP-1 to HS. In one, NRP-1 can interact with glucuronic acid containing disaccharides and the NRP-1 would, therefore, be able to bind large units containing successive NA-NAS-S-NAS domains. The alternative, in which NRP-1 binding requires iduronic acid residues, would entail NRP-1 binding a series of NAS-S-NAS units, with the NA domains forming loops away from the protein surface. In either case, an important consequence is that NRP-1 has the potential to organise the structure of large segments of HS chains in the pericellular matrix. This would be a new and important function, since the spatial organisation of HS chains in the pericellular matrix has been suggested to contribute to the control of the diffusion of morphogens and growth factors (Duchesne et al., 2012).

The interaction of shNRP-1 with heparin could only be detected by affinity chromatography and DSF, suggesting that this interaction is weak. shNRP-1 was a monomer, in contrast to Fc NRP-1, which readily oligomerises (Fig. S1 and West et al., 2005). With the c domain and L2 linker heparin binding site lacking in shNRP-1, this suggests that these sites or the possible c-domain dependent dimerisation/oligomerisation of NRP-1 may be an important mechanism for NPR-1 to bind to heparin. Thus, the existence of two distinct soluble species of NRP-1 containing either all extracellular domains (120 kDa) (Lu et al., 2009; Swendeman et al., 2008) or the four of shNRP-1 (75 kDa) (Xu et al., 2008) may have distinct functions; the latter would not be retained on the HS of the shedding cell’s pericellular matrix. Differences between membrane bound and soluble, shed species have been observed for other proteins, e.g., receptor for advanced glycation products (RAGE) and endothelial cell protein C receptor (EPCR) (Koyama, Yamamoto & Nishizawa, 2007; Molina et al., 2008). Thus, the different heparin-binding properties of the dimerised/oligomeric NRP-1 and the monomeric isoforms might drive distinct output responses of the cells.

Conclusion

The present work shows that the selectivity of NRP-1 for saccharide structures in HS is more complex than found for other heparin-binding proteins. Moreover, NRP-1 potentially may possess multiple heparin binding sites. While the direct interaction of the lysine residues identified in the selective labelling experiments clearly needs corroboration by an orthogonal approach, the potential for NPR-1 to possess multiple heparin binding sites is coherent with its preference for binding long sugar structures. This suggests that NRP-1 may have an important function in locally organising HS chains in the pericellular matrix of cells. The weak binding of shNRP-1 supports the view that the b1–b2 domain is not representative of NRP-1’s interaction with HS and that an important functional difference between NRP-1 isoforms may relate to their different ability to bind the polysaccharide.

Supplemental Information

Figure S1 S1 Schematic of the structure of the recombinant NRP-1 proteins and analysis by PAGE

(A) Human full-length NRP-1, where amino acid sequence 1-923 covers the extracellular domains (a–c), transmembrane (TM) and intracellular regions (IC). (B) The Fc linked rat NRP-1, in which two identical molecules are joined by a disulfide bridge within the Fc region. The construct contains all the extracellular domains, which is followed by a small deletion (811-828 amino acids), replaced with arginine residues, subsequent linker region originating from the original rat NRP-1 sequence, a factor Xa cleavage site (IEGRDMD), IgG1 and a His tag. (C) Soluble variant of human NRP-1, where amino acids 1-644 cover the extracellular domains, a1, a2, b1, b2, but not the c domain, and a hexhistidine (6xHis) tag. (D) Silver staining result of N-glycanase digest, performed as described in the experimental section of the reference Fc region, Fc rNRP-1 and shNRP-1, where control, mock digest and the digest results with indicated marker sizes are shown. (E) Silver staining of native PAGE resolution of 100 ng of both recombinant proteins (marker sizes are indicated).

Click here for additional data file.

Figure S2 Competition for Fc-rNPR-1 binding to heparin by heparin-derived oligosaccharides of different lengths

Competition for Fc-NRP-1 binding to the heparin derivatised surface was measured, as described in the experimental section. The relative binding values at each concentration of added sugar competitor were adjusted to 100% of Fc rNRP-1 binding alone to the immobilised heparin (measured at between 10–40 arc s). The experiment was performed three times independently (mean ± SD).

Click here for additional data file.

Figure S3 Sequence alignment of NRP-1 and NRP-1 with residues involved in binding heparin and VEGF marked

Alignment was performed with Clustal W from the initiator M and the signal sequence through to the end of L2 using sequences retrieved from Uniprot: rNRP1, rat NRP-1, accession number Q9QWJ9; hNRP1, human NRP-1, O14786; cNRP-1, chicken NRP-1, P79795; fNRP1, Xenopus laevis NRP-1; zNRP-1, zebrafish NRP-1, Q8QFX6. Residues identified in (Vander Kooi et al., 2007) as heparin binding are boxed in blue (this construct terminates at amino acid 584), lysines identified in the present work are in green and boxed in green, VEGF binding site (Geretti et al., 2007) is boxed in dark grey. Domains are overlined.

Click here for additional data file.

Figure S4 Analysis of shNRP-1 in the course of the Protect and Label procedure

(A) The panel presents the steps of analysis of shNRP-1. Lane (LD) corresponds to the sample (4 µg shNRP-1) applied to the heparin column, lane (FT) is the flow through or unbound material after applying the sample three times to the column, lanes (B1 and B2) are the two acetylation steps, lane (W1) is the post acetylation wash of the column, elution (E1) is the material eluted from the column with 2 M NaCl, lane (E2) is the same material after biotinylation (E2), and (E3) is the biotinylated material after concentration and buffer exchange. Five% (v/v) of each sampl, was analysed on SDS-PAGE and silver stained.

Click here for additional data file.

Figure S5 The spectra of the peptides analysed by MALDI-Q-TOF in the Protect and Label experiment with Fc rNRP-1

Click here for additional data file.

The authors would like to thank Dr Jonathan Popplewell from NeoSensors for DPI technical training and outstanding support and Mr Changye Sun for help with Fig. S3.

Additional Information and Declarations

Competing Interests

Author Contributions

Katarzyna A. Uniewicz is currently an employee of Qiagen GmbH.

Katarzyna A. Uniewicz conceived and designed the experiments, performed the experiments, analyzed the data, contributed reagents/materials/analysis tools, wrote the paper, prepared figures and/or tables, reviewed drafts of the paper.

Alessandro Ori performed the experiments, analyzed the data, contributed reagents/materials/analysis tools, reviewed drafts of the paper.

Yassir A. Ahmed performed the experiments, contributed reagents/materials/analysis tools, reviewed drafts of the paper.

Edwin A. Yates conceived and designed the experiments, analyzed the data, contributed reagents/materials/analysis tools, reviewed drafts of the paper.

David G. Fernig conceived and designed the experiments, analyzed the data, wrote the paper, prepared figures and/or tables, reviewed drafts of the paper.

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
