# Peer review of "Characterisation of the interaction of neuropilin-1 with heparin and a heparan sulfate mimetic library of heparin-derived sugars"

_PeerJ, doi:10.7717/peerj.461_

## Round 0.1 · original submission · Minor Revisions

Please address minor points raised by the reviewer #1.

Reviewer 1 ·

Basic reporting

In Fig.2, label A is missing. The samples used in Fig.2A and Fig.2B should be consistent. And the orientations of the labels for x axes in Fig.2B and Fig.2D are better to be the same.

Experimental design

No comments.

Validity of the findings

In line 437, the authors found “shNRP-1 interacts with heparin much more weakly
than Fc NRP1 (this work) or the b1-b2 domains in isolation.” The authors should state how similar the shNRP-1 is to rNRP-1, so that the property of the human truncated variant shNRP-1 differs from that of Rat fusion Fc- rNRP-1 is mainly because of the conformation or integrity of the proteins, not because of the difference of the protein sequences.

Additional comments

It is very interesting result that the authors identified heparin binding sites of NRP-1 by selective labelling of lysines protected upon heparin binding. As the authors have mentioned, it would need further confirmation of the binding sites. Mutations of the proposed binding sites or truncation of MAM type c domain may verify the authors’ speculations in the future. And it would be nice if the authors could show the conserved protein sequence of the mapped heparin binding sites across the species.

Reviewer 2 ·

Basic reporting

No comment

Experimental design

No comment

Validity of the findings

No comment

Additional comments

Understanding the molecular basis for Nrp’s specific binding to multiple heparin binding sites will allow us to engineer Nrp modulators that can be designed with specificity for a particular ligand. The observations made in this paper advance the search for more selective inhibitors/activators of Nrp signaling.

---

## Round 0.2 · accepted · Accept

I would like to thank you very much for the efforts you made to carefully address the reviewers' comments.